# Methane dynamics in three different Siberian water bodies under winter and summer conditions

Ingeborg Bussmann[1], Irina Fedorova[2], Bennet Juhls[3], Pier Paul Overduin[4], Matthias Winkel[5,6]

[1]Alfred Wegener Institute Helmholtz Centre for Polar and Marine Research - Helgoland, Germany

[2]St. Petersburg University, Institute of Earth Sciences, St. Petersburg, Russia

[3] Institute for Space Sciences, Department of Earth Sciences, Freie Universität Berlin, Berlin, Germany

[4]Alfred Wegener Institute Helmholtz Centre for Polar and Marine Research - Potsdam, Germany

[5] German Research Centre for Geoscience, 3.7 Geomicrobiology Group – Potsdam, Germany

[6]current address: German Research Centre for Geosciences, 3.5 Interface Geochemistry Group - Potsdam, Germany

**Correspondence to**: Ingeborg Bussmann (ingeborg.bussmann@awi.de)

Abstract

Arctic regions and their water bodies are affected by a rapidly warming climate. Arctic lakes and small ponds are known to act as an important source of atmospheric methane. However, not much is known about other types of water bodies in permafrost regions, which include major rivers and coastal bays as a transition type between freshwater and marine environments. We monitored dissolved methane concentrations in three different water bodies (Lena River, Tiksi Bay and Lake Golzovoye, Siberia, Russia) over a period of two years. Sampling was carried out under ice cover (April) and in open water (July / August). The methane oxidation (MOX) rate in water and melted ice samples from the late winter of 2017 was determined with radiotracer method and fractional turnover rates (k') from river water and melted ice cores.

In the Lena River winter methane concentrations were a quarter of the summer concentrations (8 vs 31 nmol $L^{-1}$) and mean winter MOX rate was low (0.023 nmol $L^{-1}$ $d^{-1}$). In contrast, Tiksi

Bay winter methane concentrations were 10 times higher than in summer (103 vs 13 nmol L$^{-1}$). Winter MOX rates showed a median of 0.305 nmol L$^{-1}$ d$^{-1}$. In Lake Golzovoye, median methane concentrations in winter were 40 times higher than in summer (1957 vs 49 nmol L$^{-1}$). However, MOX was much higher in the lake (2.95 nmol L$^{-1}$ d$^{-1}$) than in either the river or bay. The temperature had a strong influence on the MOX, (Q10 = 2.72 ± 0.69). In summer water temperatures ranged from 7 – 14°C, in winter from -0.7 – 1.3°C. In the ice cores a median methane concentration of 9 nM was observed, with no gradient between the ice surface and the bottom layer at the ice-water-interface. MOX in the (melted) ice cores was mostly below the detection limit. Comparing methane concentrations in the ice with the underlaying water column revealed 100 - 1000-times higher methane concentration in the water column.

The winter situation seemed to favor a methane accumulation under ice, especially in the lake with a stagnant water body. While on the other hand, in the Lena River with its flowing water no methane accumulation under ice was observed. In a changing, warming Arctic, a shorter ice cover period is predicted. In respect to our study this would imply a shortened time for methane to accumulate below the ice and a shorter time for the less efficient winter-MOX. Especially for lakes, an extended time of ice-free conditions could reduce the methane flux from the Arctic water bodies.

1 Introduction

Worldwide, the mixing ratio of methane has been increasing rapidly since 2000, from 2.1 ppb/y for the time span 2000 – 2009, to 6.6 ppb/ y for the time span 2008 – 2007 and to 6.1 ppb/y in 2017 (Saunois et al., 2020). Understanding and quantifying the global methane budget is important for assessing realistic pathways to mitigate climate change. For the 2008 – 2017-decade, global methane emissions are estimated by a top-down approach to be 576 TgCH4 /y (range 550–594, corresponding to the minimum and maximum estimates of the model ensemble) (Saunois et al., 2020). The reasons for the observed increases in atmospheric methane are unclear. A probable explanation is increased methane emissions from wetlands, both in the tropics (Nisbet et al., 2016) as well as in the Arctic (Fisher et al., 2011), and from other Arctic water bodies (Walter Anthony et al., 2016; Kohnert et al., 2018) or geological methane emissions (Kohnert et al., 2017). Especially in northern latitudes natural wetlands contribute with 59% to the northern methane emissions (Saunois et al., 2020). In the Arctic, mean atmospheric methane mixing ratio increased by 6 ppb per year from 2001 to 2017 ;

resulting in an atmospheric mole fraction of 1939 ppb in 2017 at Svalbard (Platt et al., 2018) and with a median of 1932 ppb in 2017 for Tiksi (Hydrometeorological Observatory of Tiksi, Russia). Especially the Laptev Sea, in eastern Siberia, is generally a source of methane to the atmosphere, and the sea-air flux of methane is mainly affected by increasing water temperatures (Wåhlström et al., 2016). Also, Saunois and co-authors estimated increased methane emission for freshwater systems and wetlands, but a better quantification of the emissions of different contributor (streams, rivers, lakes and ponds) is needed (Saunois et al., 2016).

Lakes are important sources of atmospheric methane on a regional to global scale (Bastviken et al., 2004; Cole et al., 2007), and their contribution is predicted to increase in response to climate change and rapidly warming waters (O'Reilly et al., 2015; Tan and Zhuang, 2015; Wik et al., 2018). Most of the methane produced in lake sediments enters the atmosphere via ebullition (Bastviken et al., 2004; Walter et al., 2007), a temperature-sensitive transport mode with high spatial and temporal heterogeneity (DelSontro et al., 2015). The role of Arctic rivers as a methane source to the shelf seas is poorly described. Some studies present rivers as strong methane sources (Morozumi et al., 2019), while other studies revealed a complex pattern of riverine methane input (Bussmann et al., 2017).

One major drawback from most of these studies is that sampling was conducted in the ice-free season, although most of the year Arctic water bodies are ice covered. Thus, the seasonal variation could not be captured within those studies. The ice cover on lakes decouples the water body from the atmosphere and the circulation changes from wind driven to thermohaline. After ice formation a stable winter stratification is set up. As there is no more external oxygen supply, enhanced anaerobic degradation leads to accumulation of methane, $H_2S$ and $NH_3$ (Leppäranta, 2015). In several lakes in Alaska and Canada, dissolved methane was highest under the ice-cover, indicating that the spring ice off period is a large source of atmospheric methane (Townsend-Small et al., 2017; Cunada et al., 2018; Serikova et al., 2019). In lakes at > 65°N the ice duration is 9 months, typically from around mid-September to mid-June (Cortés and MacIntyre, 2020). However, the hydrography of a lake is also an important factor to consider since stratification of the water column counteracts an intense gas exchange. Thus, examples are known for lakes with incomplete spring mixing and consequently a maximum gas exchange in autumn during complete mixing (Deshpande et al., 2015).

Rivers of permafrost regions are characterized by an ice season of >100 days duration between autumn freeze-up and spring ice-off; for the Lena River it is >160 days

(Shiklomanov and Lammers, 2014). Ice effects and the demobilization of liquid water result in very low discharge during winter freeze-up and runoff is lowest during late winter (Lininger and Wohl, 2019). A decrease in ice thickness of the largest Siberian Rivers during
the last 10 to 15 years (Shiklomanov and Lammers, 2014) could enhance the channel connectivity to subchannel and groundwater flow, causing an increase in winter base flow, as suggested by (Gurevich, 2009). An increased winter flow as well as increasing temperatures in the Lena River are also supported by (Tananaev et al., 2016; Yang et al., 2002; Peterson et al., 2002).

An important filter, counter-acting the methane flux into the atmosphere, is microbial methane oxidation. Methane can be oxidized under anoxic conditions close to sediment horizons where it is produced (Martinez-Cruz et al., 2017; Winkel et al., 2018) or during migration through the oxic water column to the atmosphere (Mau et al., 2017a; Bussmann et al., 2017). Under ice cover, it is important to consider methane oxidation below ice as it may
reduce the total amount of methane emitted to the atmosphere during ice-off. Active methane oxidation and a methanotrophic community has been shown for permafrost thaw ponds and lakes (Kallistova et al., 2019). Yet, the methane oxidation capacity in such lakes during ice cover with low temperatures and low oxygen concentrations is unknown. In a study covering several boreal lakes, methane oxidation was restricted to three lakes, where the phosphate
concentrations were highest (Denfeld et al., 2016). Rates of methane oxidation during the winter have been found to be much lower than summer rates, yet, there is no clear consensus on the factors limiting methane oxidation in winter (Ricão Canelhas et al., 2016). Beside oxygen concentration, the geological background (i.e. yedoma type permafrost lakes versus non-yedoma type lakes) also had a significant impact on the methane oxidation rate
(Martinez-Cruz et al., 2015).

Our study tests the hypothesis that winter ice blocks methane emissions, leading to the accumulation of methane in the underlying water bodies. By studying hydrographically different water bodies (lake, river and sea), we expect insights into the influence of water column dynamics on methane accumulation to result. In addition, we measure methane
oxidation rates in the water column and in melted ice to assess oxidation as a potential sink.

2 Study area
This study was conducted on the southern coast of Bykovsky Peninsula in northeast Siberia,
Russia (Figure 1). Thermokarst lakes in that area commonly originated in the early Holocene when surficial permafrost started to thaw, leading to accumulation of lake sediments with

organic contents of about 5-30 % (Biskaborn et al., 2016; Schleusner et al., 2015). Thermokarst lakes in the Lena River Delta seem to be ice free a little later after the coastal ice break up, depending strongly on the air temperature in the individual year (B. Juhls unpublished data).

Offshore of the Bykovsky Peninsula, part of the Yedoma Ice Complex is submerged and subsea permafrost is currently degrading. The coastline erodes at mean rates of between 0.5 and 2 m per year and can intersect inland water bodies, draining them or leading to the formation of thermokarst lagoons (Lantuit et al., 2011). The sea ice season in Tiksi Bay and the Buor-Khaya Bay typically starts in late September / early October and ends in beginning / mid-July (Angelopoulos et al., 2019; Janout et al., 2020). Due to its isolation behind Muostakh Island and Cape Muostakh, sea ice tends to be preserved longer than in the Central Laptev Sea resulting in approx. 96 days of open water (Günther et al., 2015).

The Lena River has a mean annual discharge of 581 km$^3$/year. It is the 2nd largest Arctic river by annual discharge and the 6th largest globally. There are no dams on the main-stem of the Lena, but there is a dam on the Viluy River, one of the Lena's main tributaries (Holmes et al., 2013). The Lena's watershed is 2.6 million km$^2$, of which 70.5% is underlain by continuous permafrost (Juhls et al., 2020). Most of the water is discharged during end of May and beginning of June when the ice in the rivers breaks up, but the Laptev Sea is still covered by sea ice (Holmes et al., 2012). The main Lena River branches enter the Buor-Khaya Bay through the northern and eastern part of the delta and through the Bykovskaya Channel with 20–25% of the Lena River water discharge (Charkin et al., 2011). The further distribution of the river water in Buor-Khaya Bay is mainly driven by the atmospheric systems of the cyclonic or anticyclonic Arctic circulation (Thibodeau et al., 2014; Wegner et al., 2013).

## 3 Material and Methods

### 3.1 Study sites

In the course of several expeditions to the Lena Delta (Siberia, Russia), we were able to repeatedly sample the same locations in winter and summer over the years (Table 1).

In September 2016, water samples were taken in the Bykovsky Channel and mouth of the Lena River (Overduin et al., 2017). In April 2017 with ice cover on the water bodies, ice cores were taken at Lake Golzovoye, the Lena River and Tiksi Bay, as well as the water below the ice sampled. Lake Golzovoye is an oval-shaped thermokarst lake about 0.5 km in

diameter with a maximal depth of 10 m, surrounded by Yedoma uplands at various stages of degradation and with no ice grounding in its center (Spangenberg, 2018; Strauss et al., 2018). Tiksi Bay is a shallow brackish bay at the southern end of Buor Khaya Bay, but still strongly influenced by the Lena River outflow. The water column is usually stratified, with a colder,

more saline water underlying the brackish surface layer (Overduin et al., 2015). The water of the Lena River was sampled near Samoylov island. In July and August 2017, we sampled the same locations (Lena River near Samoylov, Tiksi Bay and Lake Golzovoye) with open water (Strauss et al., 2018). The transect to the "outer" Tiksi Bay has been investigated repeatedly over the previous years (Bussmann, 2013; Bussmann et al., 2017). In April 2018, again under

ice samples were taken from the Lena River (Kruse et al., 2019).

*Table 1: Locations and sampling dates of water samples and ice cores for dissolved methane (M-conc) and methane oxidation rates (MOX).*

| Location | Sampling date | season | Water temperature (°C) | Number of samples | sampling |
|---|---|---|---|---|---|
| Lena River | 7. – 8.9.2016 | Summer | 8.5 – 10.3 | 22 | M-conc |
| Lena River | 19. – 22.4.2017 | Winter | 1.3 | 6 | M-conc, MOX |
| Lena River | 29.7.2017 | Summer | 13 - 14 | 13 | M-conc |
| Lena River | 21. – 24.4.2018 | Winter | -0.7 | 10 | M-conc |
| Lake Golzovoye | 7. – 8.4.2017 | Winter | 0.3 | 8 | M-conc |
| Lake Golzovoye | 7. – 8.4.2017 | Ice cores | | 5 | M-conc, MOX |
| Lake Golzovoye | 5.8.2017 | Summer | 7-10 | 5 | M-conc |
| Tiksi Bay | 10.4.2017 | Winter | 0.3 | 2 | M-conc |
| Tiksi Bay | 10.4.2017 | Ice cores | | 4 | M-conc, MOX |
| Tiksi Bay | 15.7 – 5.8.2017 | Summer | 4 - 11 | 6 | M-conc |

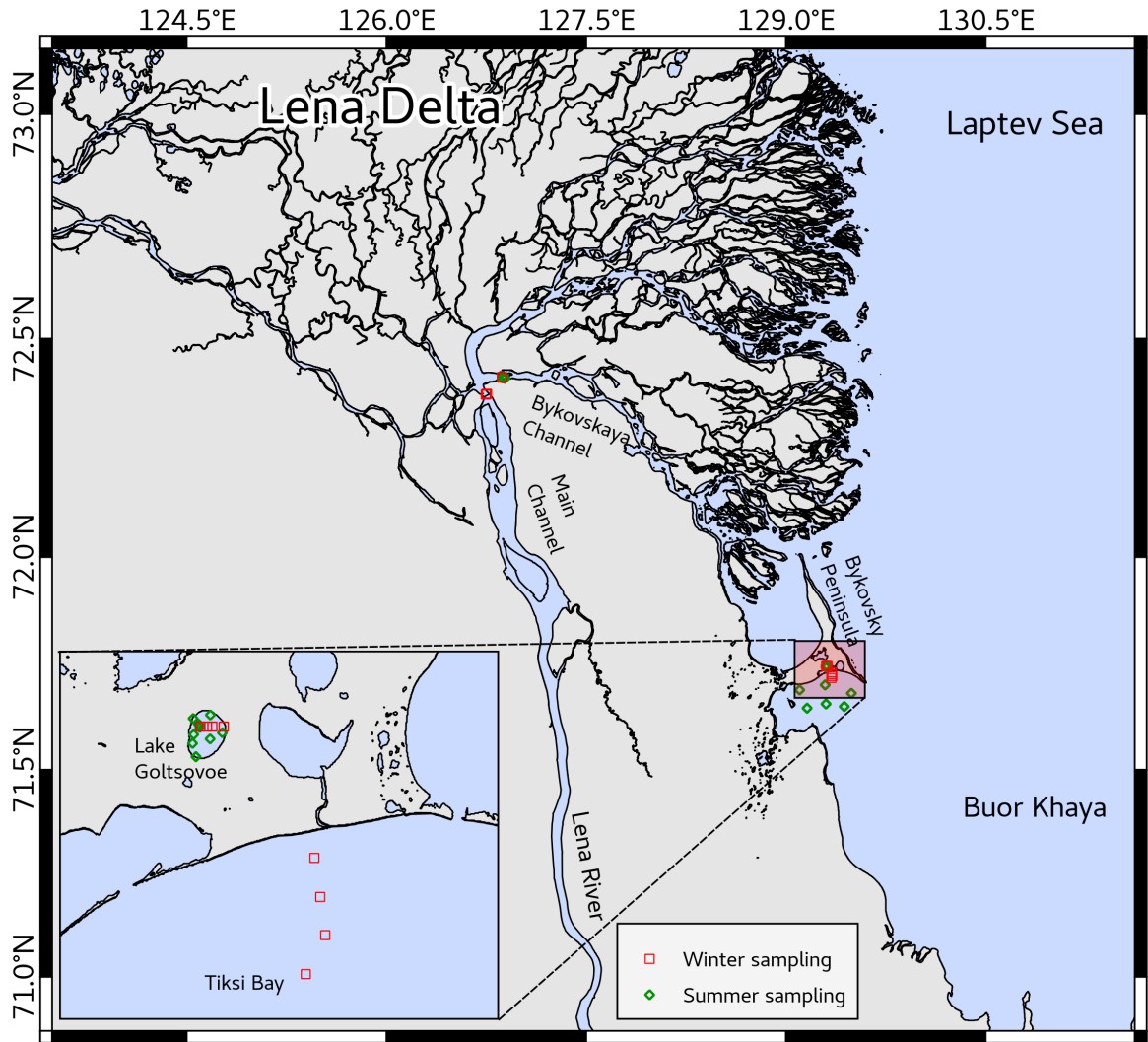

*Figure 1: Map of the study area in the Lena River and Buor Khaya Bay (Siberia, Russia.) The inlet shows details of the sampling at Lake Golzovoye. Sampling locations in winter are shown in red, summer sampling locations in green. Created by Bennet Juhls.*

### 3.2 Water sampling for methane analysis

In winter, water samples at the ice – water interface were taken with a 1 L water sampler (Uwitec Austria), transferred to 0.5 L Nalgene bottles. In the field camp, the water was transferred with a 60 ml syringe into glass bottles, closed with butyl stoppers and crimps,

poisoned with 0.2 ml 25% $H_2SO_4$ and stored upside down at 4 °C. In summer the water samples were directly transferred from the water sampler into the glass bottles, and further processed as described above.

In winter 2017, due to problems of freezing, 40 ml of sample water was shaken for 2 min with 20 ml nitrogen in a 60 ml syringe. This headspace was then transferred into glass

bottles filled with a saturated NaCl solution. Comparative measurements showed no significant difference between these two sampling strategies (Triputra, 2018). This dataset has been published already https://doi.pangaea.de/10.1594/PANGAEA.905776 .

### 3.3 Ice cores

In winter 2017, in addition to water sampling we also investigated ice cores for their methane content and methane oxidation rates. For each station three ice cores were drilled with a Kovacs Mark II ice coring system (9 cm diameter). One core was used to measure the in-situ temperature and back-up, one core was used for methane analysis, the third core was drilled for later molecular analyses. For determining the methane oxidation rates and later methane concentrations the ice cores were processed at the Research Station Samoylov Island. The top 10 cm, a 10 cm mid-section and three 10 cm-sections at the bottom of the core were cut off and transferred to special PVDF gas sampling bags (Keikaventures). The remaining parts of the cores were kept frozen. The bags were evacuated and the cores melted within approx. 5 h in a water bath at 8 °C. The sample was distributed with a 60 ml syringe to 3 glass bottles for determination of MOX and 1 bottle (which was poisoned immediately with $H_2SO_4$) for analysis of methane concentration (Strauss et al., 2018).

### 3.4 Methane analysis

Methane concentrations were determined via head-space method by adding 20 ml of $N_2$ as headspace and vigorous shaking for 2 min. Headspace methane concentrations were analyzed in the home laboratory with a gas chromatograph (GC 2014, Shimadzu) equipped with a flame ionization detector and a Shinycarbon column (Restek, USA). The temperatures of the oven, the injector, and detector were 100, 120, and 160 °C, respectively. The carrier gas ($N_2$) flow was 20 mL $min^{-1}$, with 40 mL $min^{-1}$ and 400 mL $min^{-1}$ synthetic air. Gas standards (Air Liquide) with methane concentrations of 10 and 100 ppm were used for calibration. The calculation of the methane concentration was performed according to (Magen et al., 2014). The precision of the calibration line was $r^2 = 0.99$ and the reproducibility of the samples was < 5%.

### 3.5 Methane oxidation rate (MOX)

The MOX rate was determined by adding radioactive, tritiated methane to triplicate samples (Bussmann et al., 2015). The principle of the MOX rate determination is based on the ratio of produced tritiated hydrogen from the added tritiated methane. This ratio corrected for the incubation time gives the fractional turnover rate (k' $[d^{-1}]$). To obtain the MOX rate, k' is multiplied with the in-situ methane concentration. Radioactive, tritiated methane (0.1 ml of $^3$H-methane, 2 kBq $ml^{-1}$, American Radiolabeled Chemicals) was added to triplicate samples. Samples were incubated for 60 h at 1 °C, in the dark. Incubation was stopped by adding 0.2

ml of 25% $H_2SO_4$. Abiotic controls were poisoned before adding the tracer. Radioactivity was determined with a liquid scintillation counter (Triathler, Finnland) and Ultima Gold (Perkin Elmer) as a scintillation cocktail. For MOX, the limit of detection was calculated as described in Bussmann et al. (2015) and was determined to be 0.009 nmol $L^{-1}$ $d^{-1}$ for this data set.

In a set of experiments, we also assessed the influence of temperature on the MOX rate. Water samples (Main Channel, April 2017) that were incubated at temperatures from 1; 4; 7 and 10 °C show the temperature curve of the MOX reaction. At a previous cruise off Svalbard identical experiments were performed (Mau et al., 2017a) with water from stations HE449-CTD-2, 10, 33 and 37 incubated at temperatures from 0; 4; 8; 13 and 20 °C. We determined the Q10-factor, which indicates the temperature dependence of a biological process according to (Raven and Geider, 1988):

$$Q10 = \exp(-10 * \frac{m}{Tis^2})$$

where Tis is the in-situ temperature and "m" the slope of the regression line of the Arrhenius plot (the inverse of the absolute temperature vs. the natural logarithm of the MOX rate).

## 3.6 Hydrochemistry and hydrology

Profiles of water temperature and conductivity were measured with a Cast-away CTD (SonTek) in summer 2016, and in winter and summer 2017. Water depth measurements were made with an echo-sounding device (Garmin), every 10 m along the profile. Water velocity was measured in three horizons at each vertical profile: 0.2H; 0.6H; 0.8H, with H = total water depth) using a hydrological speed recorder (GR-21). Water discharges were calculated according to the recommendations for Russian hydrometeorological stations (Fedorova et al., 2015).

# 4 Results

## 4.1 Methane concentrations and hydrochemistry

In the Lena River in summer 2016, median methane concentrations were 37 nM (n = 21) in surface water, and significantly higher than the winter concentrations of 2017 (8 nM, Wilcoxon test, p = 0.0004, Figure 2). Comparing the summer and winter concentrations for 2017/18 also showed significantly higher values in summer with a median of 25 nM versus 10

nM in winter (Wilcoxon test, p = 0.0009, Figure 2). Pooling the complete data set into winter and summer data for the Lena River showed that summer concentrations were significantly higher than winter concentrations (median of 31 nM versus 8 nM, p < 0.001). In summer 2016, water temperature ranged from 8.5 – 10.3 °C, electric conductivity from 135 – 185 $\mu$S cm$^{-1}$ and a pH-value around 6 that showed almost no variation in the water columns of 8.3 to 10 m (M. Winkler, unpublished data).

In winter 2017, the water column under the ice of the Lena River was about 1.3 °C cold with a conductivity of 275 $\mu$S cm$^{-1}$; no stratification was evident. The water was flowing at a speed of 0.043 m sec$^{-1}$ under the ice and with 0.1 m sec$^{-1}$ above the bottom (0.85 H) (Strauss et al., 2018) chapter 2.5. In the Main Channel of the Lena River the maximal water depth in winter 2017 was 25.3 m, versus 28 m in summer at the same point. In winter the water velocity in the channels is much lower than in summer: in April 2017 it was 0.18 m/sec in the Bykovskaya channel but reached 1.53 m/sec in summer 2016.

In summer 2017, the water discharge in Bykovskaya channel was 5,313.4 m$^3$/sec. Water temperature ranged from 13 – 14 °C with no changes with water depth, indicating a full mixing (Strauss et al., 2018) chapter 2.8. In Winter 2018, the water column under the ice of the Lena River was about -0.7 °C cold with a median conductivity of 465 $\mu$S cm$^{-1}$ and a median pH of 7.3. In Bykowski Channel oxygen saturation had a median value of 51%. No stratification of the water column was evident; based on water temperature and conductivity (Kruse et al., 2019). These findings agree to the Lena River monitoring observations by (Juhls et al., 2020).

In Lake Golzovoye, in summer 2017, the median methane concentration was 49 nM (n = 5), while in winter concentrations were about 40-times higher with a median of 1957 nM (n = 8, Wilcoxon test, p = 0.002, Figure 2). In winter, the water temperature of the lake was cold at the surface (median 0.3 °C in top 1 m) and warmer (2 °C) at the bottom with a thermocline at 2 meters. In summer, the surface water was heated to 7 – 10 °C, while the bottom water only warmed to 4 – 5 °C.

In Tiksi Bay and in summer 2017, the median methane concentration was 14 nM (n = 6). In contrast, in winter concentrations were about 7-times higher with a median of 103 nM (n = 2). Due to low sample numbers, no statistical test was possible. In summer 2017, water temperature ranged from 4.3 – 10.9 °C, electric conductivity from 20 – 6200 $\mu$S cm$^{-1}$. In winter, bottom water had a temperature of 3.3 °C and 0.3 °C at the top 1 m, with a thermocline at 1 – 2 meters.

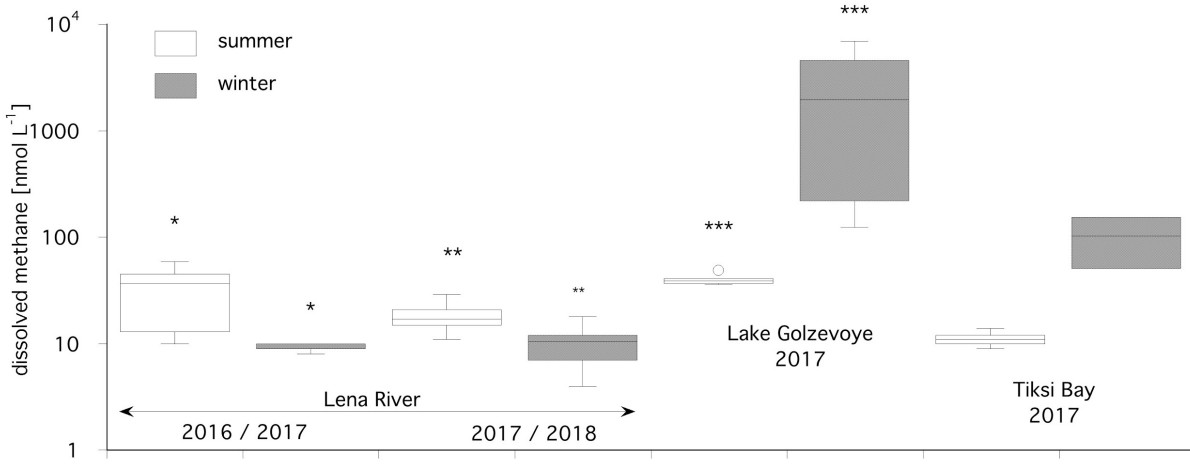

*Figure 2: Median summer and winter methane concentrations at the Lena River, Lake Golzovoye and Tiksi Bay in the years 2016 – 2018. Bright columns indicate summer values, dark columns indicate winter values. The Asterisks indicate significant differences between summer and winter data..*

## 4.2 Ice cores

Ice cores were taken at Lake Golzovoye and in Tiksi Bay, whereas no ice core data are available for the Lena River itself (Strauss et al., 2018). Methane concentrations in the ice cores of Lake Golzovoye and Tiksi Bay were rather low (both with a median of 9 nM). No depth gradients from the ice surface, middles section and the three lower most sections were evident. On a close up for the bottom layers, there was a slight increase of methane towards the ice-water interface for the ice cores from Lake Golzovoye, but not from Tiksi Bay. Figure 3 shows the median methane concentrations in the ice cores and in the water from the ice-water interface. Water column concentrations were 11times and 109times higher than in the ice cores, for Tiksi Bay and Lake Golzovoye, respectively, with a median of 102 nM and 985 nM (Bussmann and Fedorova, 2019). However, in one core of Lake Golzovoye (core #24), concentrations were orders of magnitude higher throughout the core (854 – 11091 nM) and 6954 nM in the water below**Fehler! Verweisquelle konnte nicht gefunden werden.** (Figure 3).

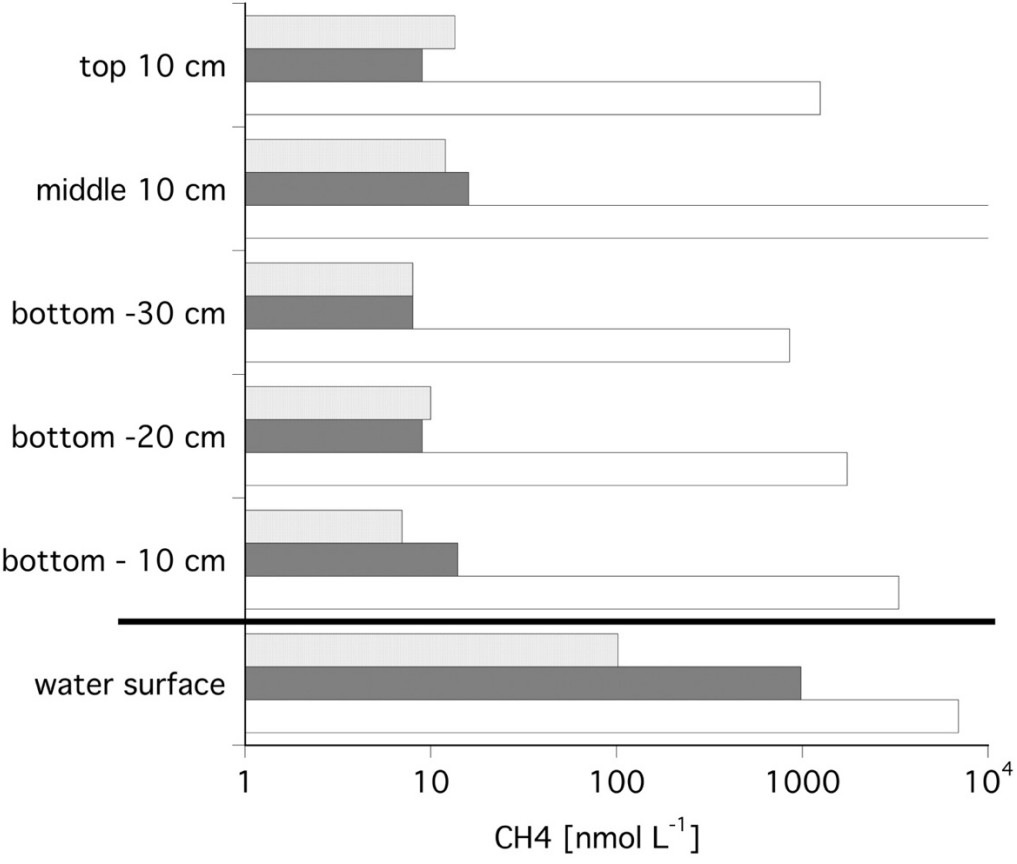


*Figure 3: Median methane concentrations in the ice cores and in the water below the ice; Date from Tiksi Bay are shown in light grey, data for Lake Golzovoye in dark grey, with core #24 shown separately in white. Note the logarithmic scale.*

**4.3 Methane oxidation rates (MOX)**

Methane oxidation rates were determined in the melted water from the ice cores from the different locations and in water from the Lena River. Due to logistic restraints at the field sites, no direct measurements of MOX in the waters of Lake Golzovoye and Tiksi Bay was possible. In the first step we determined the fractional turnover rate k'. In the ice cores, k' was

very patchy distributed and 73% of all samples were below the detection limit. In all positive samples, the value for k' was rather stable with a median of 0.003 per day (n = 31). In the water samples below the ice, the fractional turnover rate k' was never below the detection limit. The median k' determined for all water samples (Lena River) was also 0.003 (n = 14). To calculate the MOX for all water samples, we multiplied the median k' of 0.003 obtained

from Lena River and ice cores with the respective in-situ methane concentration of the water columns of Lake Golzovoye, Tiksi Bay and Lena River.

Highest MOX rates were found in the water below station #24 in Lake Golzovoye (20.36 nmol L$^{-1}$ d$^{-1}$), where also very high methane concentrations were observed (Figure 4). The median MOX in the other water samples of Lake Golzovoye was about one order of

magnitude lower (2.95 nmol L$^{-1}$ d$^{-1}$). In Tiksi Bay and in the Lena River MOX rates decreased by further orders of magnitude (median of 0.305 and 0.023 nmol L$^{-1}$ d$^{-1}$, Figure 4).

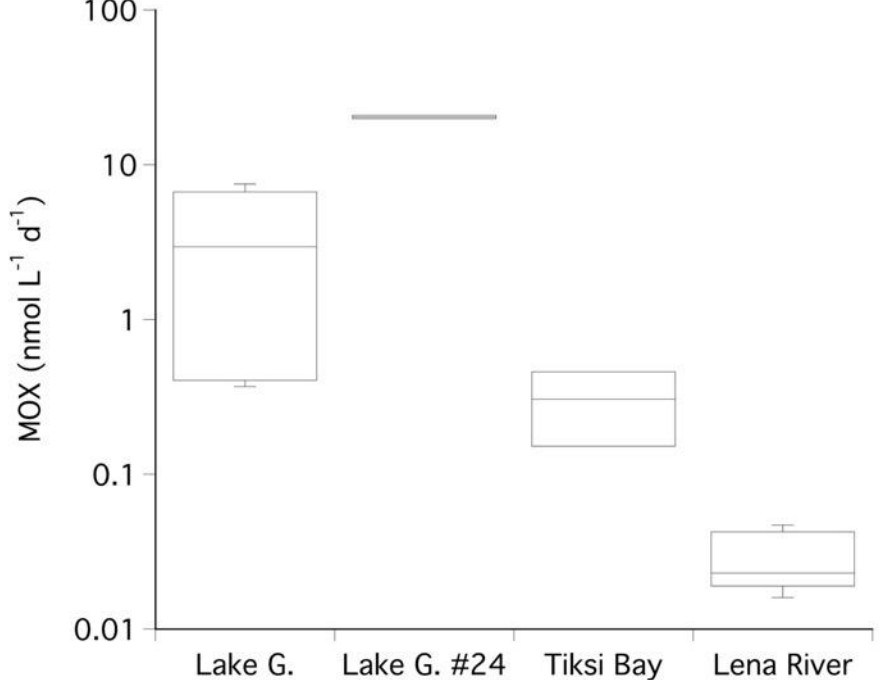

Figure 4: Box plot of the calculated methane oxidation rates (MOX) in water under ice cover at Lake Golzovoye, at the
location of ice core #24, Tiksi Bay and the Lena River. Note the logarithmic scale.

## 4.4 Temperature influence on MOX

To assess the influence of temperature on the MOX, we incubated water samples at different temperatures and determined their MOX rate. As expected, with increasing temperature the
MOX rate also increased (Figure 5). The Q10 calculated for these water samples from the Lena River was 2.72 ± 0.69. For polar and marine waters off Svalbard a Q10 of 2.99 ± 0.86 was calculated.

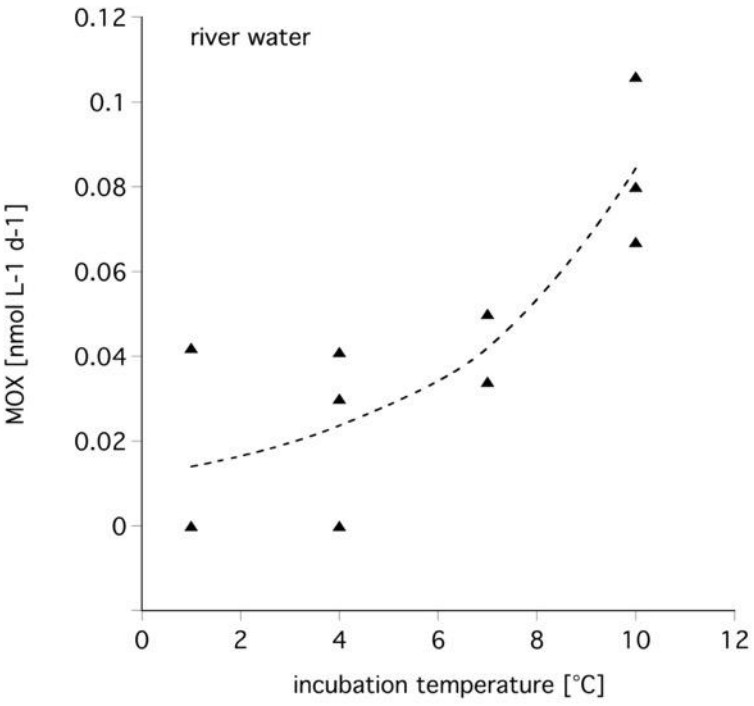

     *Figure 5: Influence of incubation temperature on methane oxidation rate for Lena River water, winter 2017.*

## 5 Discussion

### 5.1 Methane concentrations and hydrochemistry

In this study we compared the methane concentration under ice cover (winter) with open
water situations (summer) in three different water bodies.

In winter the methane concentrations in the Lena River were 4times lower than in
summer (Table 2). The Lena River displays a reduced but still substantial water flow /
discharge under ice cover. In 2017, the discharge in winter (March and April with 2.830 and
2.185 $m^3$ $sec^{-1}$) was about 10-times lower than in summer (July and August with 29.800 and
26.000 $m^3$ $sec^{-1}$, (Shiklomanov et al., 2018).

In winter there are only a few possible sources of methane left. The surrounding soils
of the drainage basins are all frozen. The ground below the main channels of the Lena is still
unfrozen (Fedorova et al., 2019), however the sediment consists of coarse grain sizes and is
poor in organic material (Rivera et al., 2006) and we do not expect any in situ methane
production. Especially in the Lena River a substantial amount of POC originates from
thermokarst-induced, abrupt collapse of Pleistocene Ice Complex deposits. These events
mainly occur in late summer, the signal however, is still visible in winter (Wild et al., 2019).
With these collapses also, methane could be imported to the river. So, we could only detect
low methane concentrations in winter. In contrast, in summer, the active layer soils from the

drainage basin allow for several sources of methane and thus increased methane concentrations in summer. Also, methane could be transported from the southern Lena catchment towards our study area, as is suggested for particulate organic matter (Winterfeld et al., 2015). At least during the warm season, methane production from (temperate) river sediments is possible (Bednařík et al., 2019). About the river-ice on the Lena there is no information yet available.

In Tiksi Bay, we observed a 7-times increase of dissolved methane under ice cover, compared to open water conditions (Table 2). Tiksi Bay is part of Buor Khaya Bay and via the central Laptev Sea perennially connected to the Arctic Ocean. Not much is known about tidal surge or water movement in Tiksi Bay under ice cover. It is anticipated that the ice cover on Tiksi Bay will lead to a decrease in tidal amplitudes and velocities (Fofonova et al., 2014). The structure of ice formation in Tiksi Bay also suggest that even in winter it is still an open system connected to the outer bay (Spangenberg et al., submitted). Sources of methane could be through diffusion of methane from the underlying sediment (Bussmann et al., 2017), where methane is produced by the degradation of organic material. However, as aerobic methane oxidation in the water column is impaired by low temperatures, methane concentrations in water increase.

In Lake Golzovoye, dissolved methane concentrations increased by 40-times from summer to winter (Table 2). Lake Golzovoye is an isolated freshwater lake with presumably only a weak thermohaline circulation (Leppäranta, 2015; Spangenberg et al., submitted). A similar, seasonal pattern of methane among lake waters of the Mackenzie Delta has been observed, ranging from very high concentrations at the end of winter beneath lake-ice (> 200.000 nM) to considerably lower concentrations during open water, particularly by late summer (approx. 1000 nM) (Cunada et al., 2018). For Yedoma lakes in Alaska, the same pattern was observed with high methane concentrations in winter, and a decrease of more than one order of magnitude in summer, (even though the actual concentrations > 900,000 nM were much higher than in this study) (Martinez-Cruz et al., 2015; Sepulveda-Jauregui et al., 2015). The same pattern is reported for boreal Swedish lakes (Denfeld et al., 2018). The source of methane in the water column of Lake Golzovoye is presumably the sediment, where high concentrations and active methane production, sulfate reduction and anaerobic methane oxidation have been observed in winter 2017 (S. Liebner, C. Knoblauch, unpublished data).

The role of water velocity and water column mixing is not clear, but our data suggest that in a stagnant water body such as a lake more methane accumulates under ice than in a water body with running water (river). Water column turbulent diffusivity has a major

influence on the methane cycle, where higher turbulence potentially leads to greater
       proportion of methane being oxidized and lower turbulence leads to a greater proportion
       being stored (Vachon et al., 2019).

**5.2 Ice cores**

The median methane concentration of all ice cores for Lake Golzovoye and Tiksi Bay was 9
       nM, which was supersaturated compared to atmospheric concentrations for which equilibrium
       concentration would be 5 nM. More details on the ice formation in the different water bodies
       are given in (Spangenberg et al., submitted). Compared to the methane concentrations in the
       water, the concentrations in the ice were 1 - 2 orders of magnitude lower (Table 2). This
means that, in terms of methane, a complete separation of the water body from the
       atmosphere can be assumed. As mentioned earlier in this study and in (Spangenberg et al.,
       submitted), core #24 in Lake Golzovoye had much higher methane concentrations throughout
       the core and visible inclusions of (methane) bubbles. We assume that core #24 was located
       above an active ebullition site, which might have slowed ice formation and prolonged direct
methane release to the atmosphere.

           In the ice itself, 28% of the samples showed methane oxidation capability. During ice
       formation most free-living bacteria are lost from the liquid phase through incorporation into
       the ice, while bacterial aggregates remain in the water (Santibáñez et al., 2019). In an
       experimental setup, (Wilson et al., 2012) show that multiple freeze-thaw cycles in water from
freshwater lakes reduce the total bacterial cell number at least 100,000-fold. In addition,
       methanotrophic bacteria are particular sensitive to freezing and thawing (Green and
       Woodford, 1992; Hoefman et al., 2012). These findings could explain the reduced activity of
       methanotrophic bacteria within the ice cores.

           Also, we did not detect any discolorations or other indications of photosynthesis or
other biological processes in the bottom layer of the ice cores. Thus, we conclude that the
       ecosystem of freshwater ice and its lower margin do not reach the richness observed in polar
       sea ice (Leppäranta, 2015).

**5.3 Methane oxidation rates (MOX)**

In this study we determined the methane oxidation rate with tritiated methane as tracer. The
       advantage of the tracer injection method is that natural low concentrations are hardly altered
       and thus we assume that our values are close to the actual rates. The fractional turnover rate k'
       was determined in ice cores from the lake and Tiksi Bay, and in river water, but not for water
       samples from the lake and Tiksi Bay. Within these locations k' was evenly distributed.

However, k' may vary between different environments (river, lake and brackish water) as well as between ice cores and underlying water. The fractional turnover rate is influenced by temperature, methane and oxygen concentrations (Steinle et al., 2017).Temperature was low at all locations and should not have a big impact on k'. For methane concentrations ranging from 6 – 800 nM, k' was independent from the methane concentration. Studies from Mau et

al 2017 and Steinle et al 2017 support the fact that the k' to methane relation does not necessarily apply. However, it cannot be excluded that at the very high methane concentration in Lake Golzovoye the real k' may have been larger. Thus, at very high methane concentrations our estimations of MOX would be an underestimation of the real rates and real k'. For all other samples, we suppose that the application of one k' to all samples is the best

possible assumption.

Our data span three orders of magnitude, ranging from 0.02 nmol $L^{-1}$ $d^{-1}$ in the Lena River, 0.31 in Tiksi Bay and 2.95 nmol $L^{-1}$ $d^{-1}$ in Lake Golzovoye. Another polar study finds a MOX of 0.004–1.09 mg C $m^{-3}$ $d^{-1}$ ( = 0.33 – 91 nmol $L^{-1}$ $d^{-1}$ ) at the water-ice interface of a Swedish lake (Ricão Canelhas et al., 2016). Also (Bastviken et al., 2002) report a MOX of

similar range (0.001–39 mg C $m^{-3}$ $d^{-1}$ = 0.08 – 3250 nmol $L^{-1}$ $d^{-1)}$. However, the methods of determining MOX were quite different, and more of a potential rate. Another study in Yedoma lakes in Alaska also reports higher MOX (0.03 – 0.28 mg methane $L^{-1}$ $d^{-1}$ = 1875 – 17500 nmol $L^{-1}$ $d^{-1}$) with a kinetic approach to determine MOX (Martinez-Cruz et al., 2015). In marine polar waters off Svalbard MOX was determined with the same tracer method and

ranged from 1.6 – 2.2 nmol $L^{-1}$ $d^{-1}$ in summer (Mau et al., 2017b). Thus, our data are within the very low range of reported MOX rates in polar regions, probably due to methodological differences.

In previous years we determined MOX in the study area during summer, applying the

same method as in this study. Therefore, we can approach a seasonal comparison (winter vs. summer), assuming interannual variability is negligible and neglecting spring and autumn mixing. To estimate the importance of ice cover on the overall MOX, we assume an ice coverage of 270 days for Lake Golzovoye and Tiksi Bay (Cortés and MacIntyre, 2020) and 160 days for the Lena River (Shiklomanov and Lammers, 2014). By multiplying the

respective winter and summer MOX with the days of ice cover and days of open water, we can calculate the amount of methane oxidized during ice cover versus ice off time.

For the Lena river and permafrost lakes we compare our winter data with summer data obtained in July 2012 (Osudar et al., 2016). For the Lena River, with a median MOX of 22.8

nmol L$^{-1}$ d$^{-1}$ (n = 8), the summer MOX was about 3 orders of magnitude higher than in winter (0.023 nmol L$^{-1}$ d$^{-1}$). The amount of methane oxidized during ice off (4674 nmol/L) was about 1270-times more than the amount oxidized during ice cover (4 nmol/L).

For MOX in lakes, summer rates from small lakes near Research Station Samoylov Island were 36-times higher (median 107 nmol L$^{-1}$ d$^{-1}$, n = 6) than winter rates (2.95 nmol L$^{-1}$ d$^{-1}$). Also, for Yedoma lakes in Alaska summer values of MOX were about 10-times higher than winter values (Martinez-Cruz et al., 2015). The amount of methane oxidized during open water (10165 nmol/L) was about 13-times more than the amount oxidized during ice cover (797 nmol/L).

For Tiksi Bay there are also summer values of MOX available (Bussmann et al., 2017). However, with a median summer rate of 0.419 nmol L$^{-1}$ d$^{-1}$ for surface, riverine water there is little difference when compared to our winter data (0.31 nmol L$^{-1}$ d$^{-1}$, Table 2). The amount of methane oxidized during open water (40 nmol/L) was about 2-times less than the amount oxidized during ice cover (84 nmol/L).

*Table 2: Comparison of methane concentration in water and ice, as well as methane oxidation rates (MOX) at different sites and different seasons.*

| Location | [CH$_4$]winter / [CH$_4$]summer* | Under ice accumulation** | Water velocity winter | [MOX] winter | MOX$_{during\ ice\ off}$ / MOX$_{during\ ice\ coverage}$ *** |
|---|---|---|---|---|---|
| Lena River | S > W, x 5 | n.a§ | strong | Low, S >> W | 4674 / 4 nmol/L  1270 |
| Lake Golzovoye | W > S, x 40 | Wa >> I, x 109 | minor | High, S > W | 10165 / 797 nmol/L  13 |
| Tiksi Bay | W > S, x 7 | Wa > I, x 11 | medium | Medium S = W | 40 / 84 nmol/L  0.5 |

§ n.a. not available

*comparing dissolved methane concentrations in water in winter and summer

**comparing methane concentration in lowest ice core layer with underlying water

*** MOX winter data from this study, summer data from (Osudar et al., 2016) and (Bussmann et al., 2017); assuming 270 days of ice coverage for Lake Golzovoye and Tiksi Bay, and 160 day of ice coverage for Lena River.

There still seems to be no clear consensus on the factors limiting MOX in winter. In several boreal lakes MOX was restricted to lakes, where the phosphate concentrations were highest (Denfeld et al., 2016). Another study reports that in winter MOX is mainly controlled by the dissolved oxygen concentration, while in the summer it was controlled primarily by the methane concentration (Martinez-Cruz et al., 2015). The stratification of lakes determining the availability of methane and oxygen for the methanotrophic bacteria also strongly influences MOX (Kankaala et al., 2006; Kankaala et al., 2007).

Temperature is also an important factor affecting winter MOX. MOX is observed at temperatures of 2 °C, mostly by *Methylococcaceae* (Ricão Canelhas et al., 2016). A recent study with *in situ* concentrations in a northern temperate lake observes a $Q10$ of $2.4 \pm 1.4$ (Thottathil et al., 2019), which is a bit lower than the $Q10$ of this study with 2.7 and 2.9 for polar, fresh and marine water, respectively. In addition, there are co-correlations between temperature and methane concentration. At substrate (methane) saturation, temperature has a strong influence, while under substrate limiting conditions, temperature has a minor influence (Lofton et al., 2014; King and Adamsen, 1992). In contrast, (Thottathil et al., 2019), observed a strong temperature response for MOX across the entire range of ambient methane concentrations. Measuring MOX with tritiated methane and thus at close to *in situ* concentrations, we can compare the $Q10$ from this study with data from temperate environments. The $Q10$ for polar environments was higher than $Q10$ values obtained from temperate waters (Elbe River, Germany and North Sea, (Bussmann et al., 2015).) with 1.52 and 1.75, respectively (Figure 6). Although the substrate concentration in the temperate waters was higher, MOX from polar regions seems to react more sensitively to a temperature increase. One explanation could be that the temperature optimum of psychrotolerant methanotrophs is below 20 °C (Bale et al., 2019). Thus, the polar methanotrophs are further away from their optimum temperature and react with increased activity to temperature increases while methanotrophic bacteria from temperate waters are nearer to their optimum temperature and do not react as sensitively.

Environmental conditions between winter and summer conditions certainly differ and may also affect the population structure of methanotrophs. Some psychrophilic strains adapt to colder temperatures (20 °C versus 4 °C) by modifying their fatty acid composition (Bale et al., 2019), while others may have only limited abilities, resulting in different population structures. This is supported by a recent study revealing that wintertime Arctic bacterial communities and food webs structure change based on carbon availability (Kellogg et al., 2019).

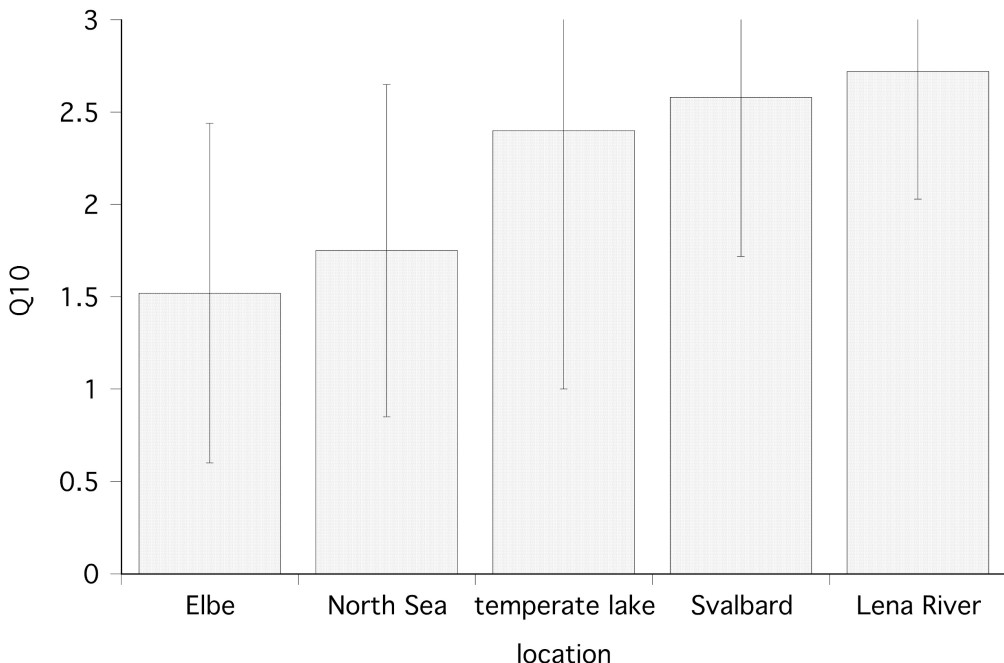

 *Figure 6: Temperature dependence (Q10) of the methane oxidation rate at different locations; shown are the mean and standard deviation, Data for Elbe and North Sea are from Bussmann et al. (2015), data for the temperate lake are from Thottathil et al. (2019).*

In this study we compared the methane inventory (concentrations) and the biological

sink (methane oxidation) of three polar aquatic environments under summer and winter

conditions. For a complete budget, the methane sources should be known, as there is methane

input from the sediment, either by diffusion or ebullition and lateral input by groundwater,

river flow or water circulation in the bay. Additional sinks for the systems are methane flux

from the water into the atmosphere and lateral output by water circulation. In the following

we apply our results on the methane cycle of the three different environments:

In the river we find higher methane concentrations and higher MOX in summer. The

low concentration in winter are probably due to low methane input from the frozen borders

and a reduced but still effective dilution of methane by the water flow (Fedorova et al., 2019).

MOX is low, and thus will not contribute to the removal of methane from the river. The ice

off on the river will probably not increase methane emissions, as only minor amounts are

accumulated under the ice.

In the lake, we observed a strong accumulation of methane in winter under ice cover.

Thus, either the methane sources are strengthened and/ or the sinks have weakened. In winter

there is an active cycle of methanogenesis and (anaerobic) methane oxidation in the sediment

(S. Liebner, unpublished data). However, we assume that this activity is the same or less than

in summer. Ebullition does occur in winter (as shown for ice core #24) and thus will lead to

locally increased methane concentrations. The methane sink, flux form the water into the atmosphere is cut off by the ice cover, thus the only remaining sink is MOX, which is reduced by low temperature and other environmental factors as discussed above. During and after ice-off, altered or weakened water column stratification will allow a mixing of the water column. This results in increased methane emission, but also to enhanced MOX as more oxygen and nutrients will become available (Utsumi et al., 1998b). In summer increased MOX and methane flux from the water lead to reduced methane concentrations in the water.

In the bay, we observed an accumulation of methane under ice and higher concentrations in winter. Thus, we assume that the sinks have weakened, with a stable or reduced methane input. Our comparison shows that MOX does not change significantly between the seasons, thus the other main sink, transport via water exchange of the bay with the shelf water is reduced during winter because of the ice cover (Fofonova et al., 2014) and direct flux from open water is reduced by the ice cover. There probably is still an input of methane from the sediments, which results than in a slight accumulation of methane. The ice off in the bay will result in increased methane emissions and also reduced methane concentrations when water circulation in the bay re-starts.

In a changing, warming Arctic, a shortened time of ice coverage on lakes and rivers is predicted (Prowse et al., 2011); (Newton and Mullan, 2020; Benson et al., 2012). This could be -6 days per 1°degree of temperature increase (Newton and Mullan, 2020). Another scenario is given by (Benson et al., 2012) with -17 days / 100 years. For the lake we observed the greatest difference between the ice-covered winter situation and open water. Especially in the lake, the duration of ice cover is important and we assume that a shorter ice cover results in less high methane concentrations under ice and subsequently in a reduced pulse of methane emissions at ice-off. In respect to our study this would imply a shortened time for methane accumulation under ice and a shortened time for the less efficient winter-MOX. For lakes this would result in increased MOX during ice off with a ratio of $MOX_{during\ ice\ off}$ / $MOX_{during\ ice\ coverage}$ of 14 and 16, versus 13 today. For rivers the same trend can be assumed with the ratio increasing from today's 1270 towards 1358 or 1539 for the two scenarios.

Thus, an extended time of ice-free conditions could reduce the methane emissions from Arctic water bodies. However, it has to be kept in mind that not much is known about the MOX during water column mixing in spring or autumn.

# 6 Conclusions

Our work on an eastern Siberian lake, river and marine bay showed that methane accumulates under ice cover during the winter and is consumed differently in the three water bodies. Our study was restricted to late winter and mid-summer, which represents two extremes of the annual cycle. Other processes during autumn mixing, ice-on, ice off are not considered.

Two main physical factors affecting the methane cycle in the water bodies under ice cover are the water velocity and the ice cover itself. In most of our ice cores no concentration gradient between the bottom of the ice cores and the top was obvious. As we could hardly detect any MOX within the ice cores, we assume that methane is not integrated into the ice during freeze up. Therefore, the ice cover seems to effectively prevent any methane flux from the highly accumulated methane concentrations in the water towards the atmosphere. In the river with running water under the ice cover, only a minor accumulation of methane was observed. In the bay with a restricted but still present water movement, dilution or mixing with other water bodies, allowed for a moderate accumulation of methane. In the small lake, we assume a stagnant water body with a subsequent accumulation of high amounts of methane.

The biotic counterpart of the observed methane accumulation is microbial methane oxidation (MOX). In most cases, MOX in summer was much higher than in winter. We observed a strong dependence of MOX on the temperature, and with in-situ temperatures of only 1 °C in winter subsequently low rates were observed. Higher methane concentrations in winter indicate the methanotrophic bacteria were not limited by substrate (methane) concentrations. However, especially in Lake Golzovoye and its stagnant water body oxygen could become a limiting factor. Other factors could be nutrient limitation or shift in the population structure.

A shortened time of ice coverage on the water bodies is predicted with increasing temperatures in the Arctic. In respect to our study this would imply a shortened time for methane to accumulate below the ice and a shorter time for the less efficient winter-MOX. Especially for lakes, an extended time of ice-free conditions could reduce the methane flux from the Arctic water bodies.

Data availability:

Data on methane concentration and MOX are available at the Pangea database.

Bussmann, I; Fedorova, I; Juhls, B et al. (2020): Dissolved methane concentrations and oxidation rates in ice cores from the Lena Delta area, 2016-2018.
https://doi.pangaea.de/10.1594/PANGAEA.920013

Bussmann, I; Fedorova, I; Juhls, B et al. (2020): Dissolved methane concentrations and oxidation rates in water samples from the Lena Delta area, 2016-2018.
https://doi.pangaea.de/10.1594/PANGAEA.919986

Author contribution:
All authors carried out field work and measurements and collected samples. IB performed the methane and MOX analyzes. IB, BJ, PPO and MW contributed to the initial and final versions of the manuscript.

Competing interests:
The authors declare that they have no conflict of interest.

Acknowledgements:

This study was part of the Helmholtz program PACES, Topic 1.3. We are thankful to the logistics department of the Alfred Wegener Institute, particularly Waldemar Schneider. Logistical support for the fieldwork was provided by the Russian Hydrographic Service
(Hydrobase Tiksi). Matthias Winkel was supported by the Helmholtz Young Investigators Group of Susanne Liebner (VH-NG-919) and further supported by the German Ministry of Education and Research by a grant to Dirk Wagner (03G0836D). We acknowledge support by the Open Access Publication Funds of Alfred-Wegener-Institut Helmholtz-Zentrum für Polar- und Meeresforschung.

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
