# Peer review of "Methane dynamics in three different Siberian water bodies under winter and summer conditions"

_Biogeosciences, 2020_

## Referee Comment (RC1) · Anonymous Referee #1 · 9 Jul 2020

The manuscript "Seasonal methane dynamics in three different Siberian water bodies" by Bussmann et al. presents measurements of methane concentrations and oxidation rates (MOX) from different samplings in an Arctic river delta, in the adjacent coastal waters and in a nearby small Arctic lake from sampling campaigns carried out during ice-free summer conditions and under ice cover in late winter. The presented data close knowledge gaps on winter time methane dynamics in Arctic waters. The authors find that $CH_4$ accumulation under ice is governed by the exchange of waters under the ice, with insignificant accumulation in the river and strong accumulation in the stagnant lake waters. Additional measuremements from ice cores did not show large $CH_4$ accumulation and no $CH_4$ gradients, and confirmed that the ice cover is an effective barrier for the $CH_4$ flux to the atmosphere.

[Figure]

The manuscript is generally well written, and the presentation of the CH4 concentration and ice core data is clear and comprehensive.

I have some issues with the part on the MOX rates, however, which need to be resolved by the authors: in addition to the methane concentrations from the different water bodies, MOX rates were determined in winter time in the Lena river. The authors use these MOX rates to determine the potential of MOX for reducing the CH4 accumulation under ice. Based on their assumptions, they conclude that the measured low MOX would lead to a rather small reduction of the CH4 inventory, and strong CH4 accumulation under ice is likely. However, this estimate is based on a number of assumptions the authors make, and I am missing a detailed chain of arguments that justifies these assumptions:

- the authors transfer the fractional MOX rate determined from the Lena river sampling to the coastal and lake waters (lines 318-320), but they miss a convincing argumentation if this isjustified. In lines 460 to 470, they discuss the influence of different environmental factors such as oxygen and phosphate concentration and temperature (lines 460-470), but these factors are not taken into account or even discussed when the MOX rates from the coastal and lake waters are calculated. Instead, the authors present the MOX rates from these two water bodies as if these were independently determined (lines 320-325, Fig.5), and not calculated from the CH4 concentrations.

- the authors conclude that the accumulation of CH4 under ice will lead to a pulse release of CH4 when the ice melts, and the low MOX in winter compared to summer conditions induce larger overall emissions from winter than from summer conditions. I think this argumentation is a bit too superficial – summer and winter conditions not only differ in the effectiveness of CH4 oxidation, but also in the fact that during summer, gas exchange is an additional sink for CH4, and that CH4 emissions are determined by the effectiveness of CH4 oxidation vs. gas exchange. To fully assess the potential of CH4 emissions from pulse release during ice-off or from continuous emissions under ice-free conditions, all sink terms would need to be carefully taken into account.

Specific comments:

Line 20: "Arctic regions and their water bodies are affected..."

Line 134: "the Bykovskaya Bay"

Line 467: "But temperature is also...": remove "but"

Lines 504-505: A 60% reduction of the methane inventory by MOX does not seem like an insignificant reduction of the CH4 inventory to me.

Lines 508-509: "we assume, that after ice-on, both parameters increase/decrease in a linear way" Could you explain this assumption? Why would this be justified?

Lines 546-550: I think this statement is somewhat speculative- I doubt that the increased MOX rates in summer than in winter is the only factor that determines CH4 emissions from the Arctic water bodies. The authors should at least mention which additional factors could influence CH4 emissions.

Line 554: The data should be submitted to PANGAEA and the data availability statement updated.
* * *

---

## Referee Comment (RC2) · Anonymous Referee #2 · 7 Sep 2020

This is a very interesting study and it is also practically difficult to conduct field research in waterbodies of Arctic regions. It is thus a timely contribution of methane cycles.

The manuscript is well organized, and the writing appears to be somehow redundant.

The major concerns are the followsing.

(1) Title. Is the term seasonal dynamic appropriate? There are only two sampling period for some rivers. A better title might be formulated such methane dynamics under contrasting ***? (2) Conclusion. The rationale behind the higher methane concentrations in winter than that in summer is not very clear for Tiksi bay and Lake Golzovoye. Please make a brief and focused discussion about the possible mechanisms. (3) Methane production potential. If these data are not available, the authors may dis-

cuss methanogenesis a little bit more. Or methane simply stored in waterbodies due to physiochemical mechsnisms? (4) Oxygen concentrations. Please provide these data as much as possible if available. Major con

Minor concerns

(1) L20. How to define "the most rapid climate warming on Earth"? (2) L22. Maybe the authors can briefly introduce the proportion of these poorly unexplored water bodies. (3) L35. It is somehow abrupt to compare with temperate environments. This is more appropriate in the review paper (4) L45. Please give concluding remarks as a summary of the key findings. (5) L55. Pls describe the range of variability (6) L65-67. This sentence seems irrelevant to the previous one. The ebullition mode and transportation from Arctic rivers to the shelf seems to be different. (7) L108. Pls write the conclusions in the abstract in line with the hypothesis. (8) L111. Why not measure the potential of methanogenesis, and how to integrate these potential in situ sink with the budget estimate of methane emission? (9) L125. The freeze-up and ice-off days can be specified for each waterbody (10) L168. How low it is below the ice? (11) L174. Is the sampling procedure the same for different rivers? (12) L195. Please describe the procedures for methane concentration measurement. For example, is there any vigorous shakingïij§ (13) L305. Figure 3 and Figure 4 can be mereged. (14) L340. Figure 5 and Figure 6 can be merged (15) (16) (17)

---

## Author Response (AR1)

Response to the editors comments
Dear Dr. Bussmann

*This is an interesting study, particularly considering the difficulty for sampling*

*Both reviewers provide critical comments which cannot be corrected by minor changes. Substantial changes appear to be needed to address adequately the questions raised by both reviewers.*

*In addition, the following are for the authors' reference by a a quick editorial review again*
*(1) Please justify the rationale why two sampling could be defined as "seasonal dynamics"*
ok, the title has been changed to "Methane dynamics in three different Siberian water bodies under winter and summer conditions"

*(2) Please describe the most rapid climate warming on Earth in a quantitative manner in the introduction section*
The increase of atmospheric methane has been specified with more recent data.
"Worldwide, the mixing ratio of methane has been increasing rapidly since 2000, from 2.1 ppm/y for the time span 2000 – 2009, to 6.6 ppb/ y for the time span 2008 – 2007 and to 6.1 ppb/y in 2017 (Saunois et al., 2020). Understanding and quantifying the global methane budget is important for assessing realistic pathways to mitigate climate change. For the 2008 – 2017 decade, global methane emissions are estimated by a top-down approach to be 576 TgCH4 /y (range 550–594, corresponding to the minimum and maximum estimates of the model ensemble) (Saunois et al., 2020)"

*(3) Line 36. Is it positive influence when compared to temperate environments*
The comparison to temperate environments has been removed from the abstract. In the text however, it can be seen that the Q10 is for polar environments was higher than for temperate environments.

*(4) In the abstract, please conclude with a more general finding if possible. Or maybe the authors could add some important implications of this study*
The following sentence has now been added to the end of the abstract
…."In a changing, warming Arctic, a shortened time of ice coverage on the water bodies is predicted. In respect to our study this would imply a shortened time for methane to accumulate below the ice and a shorter time for the less efficient winter-MOX. Especially for lakes, an extended time of ice-free conditions could reduce the methane flux from the Arctic water bodies.

*(5) L53. Pls specify the current concentration of methane in the atmosphere.*
The atmospheric methane concentration as measured in the polar region of Svalbard is given with 1939 ppb for 2017 (Platt et al 2018) and with 1932 ppb for Tiksi (Hydrometeorological Observatory of Tiksi, Russia (TIK).

*(6) L55-58. Why a better quantification of the emissions of different contributors is needed. What is the current estimate, and how uncertain it is, and the range ?*
We added the following sentence:

Understanding and quantifying the global methane budget is important for assessing realistic pathways to mitigate climate change. For the 2008–2017 decade, global methane emissions are estimated by a top-down approach to be 576 TgCH4 /y (range 550–594, corresponding to the minimum and maximum estimates of the model ensemble) [Saunois, 2020 #3085].
And
Especially in northern latitudes natural wetlands contribute with 59% to the northern methane emissions (Saunois et al., 2020)

*I would like to have your attention that simple changes will not be able to correct all the comments raised by the reviewers*

*Thank you for your submission to BG and once again I apologize for some delay of managing the ms.*

*Kind regards*

Interactive comment on "Seasonal methane dynamics in three different Siberian water bodies"

by Ingeborg Bussmann et al.

Anonymous Referee #1

The manuscript "Seasonal methane dynamics in three different Siberian water bodies" by Bussmann et al. presents measurements of methane concentrations and oxidation rates (MOX) from different samplings in an Arctic river delta, in the adjacent coastal waters and in a nearby small Arctic lake from sampling campaigns carried out during ice-free summer conditions and under ice cover in late winter. The presented data close knowledge gaps on winter time methane dynamics in Arctic waters. The authors find that CH4 accumulation under ice is governed by the exchange of waters under the ice, with insignificant accumulation in the river and strong accumulation in the stagnant lake waters. Additional measurements from ice cores did not show large CH4 accumulation and no CH4 gradients, and confirmed that the ice cover is an effective barrier for the CH4 flux to the atmosphere.

The manuscript is generally well written, and the presentation of the CH4 concentration and ice core data is clear and comprehensive.

I have some issues with the part on the MOX rates, however, which need to be resolved by the authors: in addition to the methane concentrations from the different water bodies, MOX rates were determined in winter time in the Lena river. The authors use these MOX rates to determine the potential of MOX for reducing the CH4 accumulation under ice. Based on their assumptions, they conclude that the measured low MOX would lead to a rather small reduction of the CH4 inventory, and strong CH4 accumulation under ice is likely. However, this estimate is based on a number of assumptions the authors make, and I am missing a detailed chain of arguments that justifies these assumptions:

- *the authors transfer the fractional MOX rate determined from the Lena river sampling to the coastal and lake waters (lines 318-320), but they miss a convincing argumentation if this is justified. In lines 460 to 470, they discuss the influence of different environmental factors such as oxygen and phosphate concentration and temperature (lines 460-470), but these factors are not taken into account or even discussed when the MOX rates from the coastal and lake waters are calculated. Instead, the authors present the MOX rates from these two water bodies as if these were independently determined (lines 320-325, Fig.5), and not calculated from the CH4 concentrations.*

Yes, the referee is right here, we determined k' for river water and for the ice cores from the Lake and Tiksi Bay, but k' was not determined for each single sample. This is a major drawback, but the best we could do at these difficult locations. We also removed the term "in situ MOX" and name it only MOX. This fact is stressed more now in the abstract, the result section, as well as in the discussion: "The fractional turnover rate k' was determined in ice cores from the lake and Tiksi Bay, and in river water. Within these locations k' was evenly distributed. However, k' may vary between different environments (river, lake and brackish water) as well as between ice cores and underlying water. The fractional turnover rate is influenced by temperature, methane and oxygen concentrations [Steinle, 2017 #2755]. Thus, in our calculations an error is embedded. However, as temperature was low at all locations, and the methanotrophic population of the ice cores is similar to the ones in the water below, we assume that the application of one k' to all samples is a good assumption."

*- the authors conclude that the accumulation of CH4 under ice will lead to a pulse release of CH4 when the ice melts, and the low MOX in winter compared to summer conditions induce larger overall emissions from winter than from summer conditions. I think this argumentation is a bit too superficial*
We agree that the line of discussion was too simplified. We added now a more detailed discussion on the sources and sinks of methane in an aquatic system. As can be seen in the last part of the new discussion

*– summer and winter conditions not only differ in the effectiveness of CH4 oxidation, but also in the fact that during summer, gas exchange is an additional sink for CH4, and that CH4 emissions are determined by the effectiveness of CH4 oxidation vs. gas exchange. To fully assess the potential of CH4 emissions from pulse release during ice-off or from continuous emissions under ice-free conditions, all sink terms would need to be carefully taken into account.*
We agree that the line of discussion was too simplified. We added now a more detailed discussion on the sources and sinks of methane in an aquatic system. As can be seen in the last part of the new discussion.

Specific comments:
*Line 20: "Arctic regions and their water bodies are affected..."*
Changed accordingly

*Line 134: "the Bykovskaya Bay"*
No, Buor-Kaya Bay is correct

*Line 467: "But temperature is also: : :": remove "but"*

Changed accordingly

*Lines 504-505: A 60% reduction of the methane inventory by MOX does not seem like an insignificant reduction of the CH4 inventory to me.*
This part has been removed and rewritten.

*Lines 508-509: "we assume, that after ice-on, both parameters increase/decrease in a linear way"*
*Could you explain this assumption? Why would this be justified?*
This part has been removed and rewritten.

*Lines 546-550: I think this statement is somewhat speculative- I doubt that the increased MOX rates in summer than in winter is the only factor that determines CH4 emissions from the Arctic water bodies. The authors should at least mention which additional factors could influence CH4 emissions.*
This part has been removed and rewritten.

*Line 554: The data should be submitted to PANGAEA and the data availability statement updated.*
The data are submitted in Pangaea and details are given in the revised version.

Anonymous Referee #2

This is a very interesting study and it is also practically difficult to conduct field research in waterbodies of Arctic regions. It is thus a timely contribution of methane cycles.

The manuscript is well organized, and the writing appears to be somehow redundant.

The major concerns are the following.

*(1) Title. Is the term seasonal dynamic appropriate? There are only two sampling period for some rivers. A better title might be formulated such methane dynamics under contrasting ***?*

As the editor also suggest to remove the "seasonal" , we changed the title to "Methane dynamics in three different Siberian water bodies under winter and summer conditions"

*(2) Conclusion. The rationale behind the higher methane concentrations in winter than that in summer is not very clear for Tiksi bay and Lake Golzovoye. Please make a brief and focused discussion about the possible mechanisms.*

A new part of the discussion has been added, to explain in more detail the different sinks and sources of methane in a water body. (in the last part of the discussion)

(3*) Methane production potential. If these data are not available, the authors may discuss methanogenesis a little bit more. Or methane simply stored in waterbodies due to physiochemical mechanisms?*

Colleagues have determined the methane production rate in Lake G. and a marine setting, with 0.5 - 0.2 nmolg/g/d at surface sediment. However, active AOM and SRR reduce methane concentrations at the sediment surface to about 2 mM in freshwater sediment and to < 0.1 mM in marine sediments (unpublished data from S. Liebner and C. Knoblauch). Added in Line 445ff

*(4) Oxygen concentrations. Please provide these data as much as possible if available.*

Unfortunately, there are no data on the oxygen concentrations at any of the locations.

Minor concerns

    *(1) L20. How to define "the most rapid climate warming on Earth"?*

        Changed to "affected by a rapidly warming climate"

*(2) L22. Maybe the authors can briefly introduce the proportion of these poorly unexplored water bodies.*

More details on the different water bodies are given in the introduction, . L 87 ff and L 111 ff. To elaborate this in the abstract would be too long

> (2) *L35. It is somehow abrupt to compare with temperate environments. This is more appropriate in the review paper*

The relation to temperate environments has been removed.

*(4) L45. Please give concluding remarks as a summary of the key findings.*

We have now added our concluding remarks to the abstract.

"The winter situation seemed to favor a methane accumulation under ice, especially in the lake with a stagnant water body. While on the other hand, in the Lena River with its flowing water no methane accumulation under ice was observed. In a changing, warming Arctic, a shorter ice cover period is predicted. In respect to our study this would imply a shortened time for methane to accumulate below the ice and a shorter time for the less efficient winter-MOX. Especially for lakes, an extended time of ice-free conditions could reduce the methane flux from the Arctic water bodies."

*(5) L55. Pls describe the range of variability*

This variability is mentioned in the abstract of the reference, but not further elucidated in the text. Therefore we changed our statement to "…and the sea-air flux of methane is mainly affected by increasing water temperatures (Wåhlström et al., 2016)"

*(6) L65-67. This sentence seems irrelevant to the previous one. The ebullition mode and transportation from Arctic rivers to the shelf seems to be different.*

In the sentences above we describe the role of lakes for the methane flux. In the lines 111 ff we wanted to elucidate the role of rivers as methane source to the atmosphere, not the shelf…..

*(7) L108. Pls write the conclusions in the abstract in line with the hypothesis.*

We have now added our concluding remarks to the abstract.

*(8) L111. Why not measure the potential of methanogenesis, and how to integrate these potentials in situ sink with the budget estimate of methane emission?*

We were not able to measure methanogenesis in the field, however there are unpublished data on methane production and anaerobic oxidation in the sediment of Lake Golzovoye. We mention now these unpublished data in L 445ff.

*(9) L125. The freeze-up and ice-off days can be specified for each waterbody*

We have now added additional information, for lakes in L 105ff and L151, and for Tiksi Bay in L 158.

*(10) L168. How low it is below the ice?*

Just at the interface between ice and water, changed to "In winter, water samples at the ice – water interface were taken...."

*(11) L174. Is the sampling procedure the same for different rivers?*

No, the river water has been processed in the same way.

*(12) L195. Please describe the procedures for methane concentration measurement. For example, is there any vigorous shakingüj§*

Yes, samples were shaken for 2 min, to assure equilibrium between water and gas phase. This has been added to the Result section (3.4 Methane analysis).

*(13) L305. Figure 3 and Figure 4 can be merged.*

Ok, the figures have been merged

*(14) L340. Figure 5 and Figure 6 can be merged*

No, Figure 5 shows the methane oxidation rate at the different locations, while figure 6 shows the influence of temperature on the methane oxidation rate.

[revised manuscript text omitted]

---

## Author Response (AR2)

*Dear Dr. Zhongjun Jia,*
*Thank you for your decision and comments from reviewer 1.*
*I have now addressed all points raised and hope that critics on the MOX calculation are*
*clarified to satisfy you and the reviewer.*

*Yours sincerely*

*Ingeborg Bussmann*

Dear Dr. Bussmann,

Your manuscript has been reviewed by an external expert and myself. The criticism raised
during the first round of reviews has been addressed, but still the justification of methane
oxidation rate needs to be strengthened and discussed in greater details.

Meanwhile, the figure quality can be significantly improved, particularly for Figure 1. Please
also remember that it is expected that your figure legends will be quite detailed and very
precise. In fact, from the figure title and the axis labels of a graph/table the reader should be
able to determine the question being asked, get a good idea of how the study was done, and be
able to interpret the figure without reference to the text

Yours sincerely

Zhongjun Jia

*The legend of figure 1 has been extended. Also, the legends of the other figures have been*
*checked and amended.*

**Suggestions for revision or reasons for rejection (will be published if the paper is
accepted for final publication)**

The manuscript "Methane dynamics in three different Siberian water bodies under winter and
summer conditions" by Bussmann et al. has been substantially revised by the authors and
most of my previous comments have been taken into account.

I still think that the argumentation on the MOX rates could be further strengthened, however.
The discussion of the manuscript heavily relies on the calculated MOX rates, and in my
previous review of the manuscript my main criticism was that I missed a chain of arguments
that justifies the calculation of MOX rates for all water types from the fractional MOX rates
measured in ice core and river samples.

In their answer to my review, the authors agreed that their approach has some drawbacks but
stated that due to the difficult logistics, transferring the MOX rates measured in the ice core
and river waters to the other water bodies is the best they can do. While I understand that
conducting MOX rate incubation experiments under field conditions is very challenging, I

still think the authors need to discuss the limitations of their approach in the manuscript more clearly. In lines 449-454 the authors state that transferring the MOX rates from ice core and river data to the other bodies is justified due to the fact that MOX rates are generally low at low temperatures and due to the fact that the methanotrophic population (does this refer to the community composition?) of the ice cores is similar to the one measured in the water below. While I agree that the temperature dependence of the MOX rate indicates that MOX is low at low temperatures, I am not fully convinced that other factors have sufficiently been taken into account by the authors. Particularly the influence of CH4 concentrations on MOX rates may be of importance, as CH4 concentrations in the lake and the coastal embayment were significantly higher below the ice than in the river water.

*The MOX rate is calculated by multiplying k' with the corresponding CH4 conc, thus only k' is per se independent from the CH4 conc. However, on a physiological or population level k' may well depend on the substrate concentration. We tested this for our data set: for methane concentrations ranging from 3 – 800 nM k' is independent from the CH4 concentration. Studies from Mau et al 2017 and Steinle et al 2017 support the fact that the k' to CH4 relation does not necessarily apply. However, it cannot be excluded that at the very high methane concentration as in Lake G. k' may also increase. Thus, our estimations of MOX would be an underestimation of the real rates and real k'.*

*The respective text in the discussion has been changed to:*
*In this study we determined the methane oxidation rate with tritiated methane as tracer. The advantage of the tracer injection method is that natural low concentrations are hardly altered and thus we assume that our values are close to the actual rates. The fractional turnover rate k' was determined in ice cores from the lake and Tiksi Bay, and in river water, but not for water samples from the lake and Tiksi Bay. Within these locations k' was evenly distributed. However, k' may vary between different environments (river, lake and brackish water) as well as between ice cores and underlying water. The fractional turnover rate is influenced by temperature, methane and oxygen concentrations (Steinle et al., 2017).Temperature was low at all locations and should not have a big impact on k'. For methane concentrations ranging from 6 – 800 nM, k' was independent from the methane concentration. Studies from Mau et al 2017 and Steinle et al 2017 support the fact that the k' to methane relation does not necessarily apply. However, it cannot be excluded that at the very high methane concentration in Lake Golzovoye the real k' may have been larger. Thus, at very high methane concentrations our estimations of MOX would be an underestimation of the real rates and real k'. For all other samples, we suppose that the application of one k' to all samples is the best possible assumption.*

I furthermore have some minor comments that should be taken into account:

Lines 34 and 36: replace "10-times" and "40-times" with "10 times", "40 times"
*corrected*

Line 55: I guess it should read "from 2.2 ppb/yr to..."
*corrected*

Line 67: replace "concentration" by "mole fraction"

*corrected*

Lines 67-69: CH4 emissions from water bodies (incl. lakes, rivers, coastal waters) have not been mentioned in the introduction before. How significant are these emissions in relation to other sources in the northern latitudes?
*Methane emissions from lakes and rivers are discussed in the following paragraph.*

Lines 189-193: if the samples are transferred to Nalgene bottles before filling the glass bottles, how is CH4 loss from the samples prevented?
*The Nalgene bottles were completely filled to the top and transfer to the glass bottles occurred within 3 hours of sampling. This procedure does include a certain methane loss, however handling tubings, stoppers, cannulas and glass bottles at freezing temperatures proved to be more error bound.*

Lines 313-315: "In Figure 3 the median concentrations in the … ": this formulation is somewhat strange. Please rephrase this sentence to something like "Figure 3 shows the median CH4 concentrations in the ice core and at in the water from the ice-water interface in…"
*corrected*

Lines 319-320: there seems to be something wrong with the reference that should be cited here.
*corrected*

Lines 328-329: can the authors give additional information on the CH4 concentrations in the samples that were used to determine MOX rates? Did the authors find any correlation between MOX rate and concentrations?
*As outlined above, only k' is per se independent from the methane conc., while MOX is calculated by multiplying k' with the methane concentration. Thus, we focus more on the discussion of k'. In the samples in which k' was determined, methane concentrations ranged from 3 – 800 nM and  k' was independent from the  M-conc.*

Lines 333-335: This sentence repeats the previous one.
*The latter sentence has been deleted.*

Lines 337-340: Does this mean that an overall median MOX (mean of ice cores and water samples) is transferred to the entire dataset? If MOX in 73% of the ice cores is below the detection limit would the overall median value from the ice cores not be below detection limit, too? Would it not make more sense to transfer the MOX rates from only the water samples to the other water bodies?
*As explained in the text, in all positive (ice core) samples the median k' was 0.003.  The samples below the detection limit were excluded here. The k' from positive ice cores samples and from river water samples was both 0.003.*

Line 366: "lower than in summer"
*corrected*

Lines 618-619: this sentence is a repetition of lines 578-579.

*The latter has been changed to "A shortened time of ice coverage on the water bodies is predicted with increasing temperatures in the Arctic"*

Table 1: maybe the information on samplings for the MOX rate measurements can be added here? I think it is interesting to know where and when the authors took the samples for these incubations.
*This information has been added to table 1*

Figure 3: I would suggest to swap the data from Lake Golzovoye and Tiksi Bay, to show the Lake Golzovoye data next to the data from the separate ice core from the lake.
*Changed accordingly*

---

## Author Response (AR3)

*Comments to the Author:*
*Dear Dr. Bussmann*

*Thank you for submitting the revised ms to BD.*

*I have again read the ms as both an external reviewer and editor.*

*Yes, I do agree with you and reviewer#1 that this is an important piece of findings, although I am still not 100% convinced that temperature played a key role in the remarkable difference in MOX. But I do appreciate your careful and elegant explanation.*

*So, my minor concern is that you may add the range of temperature in winter and summer in the abstract, particularly in the figure legend wherever appropriate. So the readers could get an immediate sense of the variation.*

*Yours sincerely*

*Zhongjun*

Dear Dr. Zhongjun,
thank you for all your efforts.
I have now included the range of winter and summer temperatures in the abstract. I do not think that water temperatures would fit to any of the figures, however I have included them into table 1.
Yours sincerely
Ingeborg Bussmann